# Influence of the Use of Permanent Catalytic Systems on the Flue Gases Emission from Biomass Low-Power Boilers

Błażej Gaze [1,*], Bernard Knutel [1], Mateusz Jajczyk [2], Ondřej Němček [3], Tomáš Najser [3] and Jan Kielar [3]

[1] Institute of Agricultural Engineering, Wroclaw University of Environmental and Life Sciences, 51-630 Wroclaw, Poland; bernard.knutel@upwr.edu.pl

[2] Faculty of Environmental Science and Technology, Wroclaw University of Environmental and Life Sciences, 50-363 Wroclaw, Poland; 116135@student.upwr.edu.pl

[3] Centre of Energy Utilization of Non-Traditional Energy Sources—ENET Centre, CEET, VSB—Technical University of Ostrava, 17. Listopadu 2172/15, 708 00 Ostrava, Czech Republic; ondrej.nemcek@vsb.cz (O.N.); tomas.najser@vsb.cz (T.N.); jan.kielar@vsb.cz (J.K.)

\* Correspondence: blazej.gaze@upwr.edu.pl; Tel.: +48-71-320-57-15

**Abstract:** The paper presents the research results on the use of permanent catalytic systems applied to the surface of a low-power boiler deflector. The tests were carried out on a standard 15 kW retort boiler. The boiler was powered by three types of biomass pellets (wood pellets, wheat straw pellets, and hemp expeller). In the research cycle, the influence of the catalysts on the emission of individual compounds, CO, $NO_X$, particulate matter (PM), polycyclic aromatic hydrocarbons (PAH), and volatile organic compounds (VOC) and the influence on the temperature in the combustion chamber were examined. The tests used an exhaust gas analyzer, a dust meter, a two-channel aspirator, and a laboratory gas chromatograph stand with a flame ionization detector. Four catalysts (copper, manganese, titanium, and platinum) were prepared for the analysis. Each catalyst had three variants of the active substance concentration on the ceramic support surface: 17.5 g, 35 g, 52.5 g for $CuO$, $TiO_2$, $MnO_2$, and, respectively, 0.05 g, 0.1 g, and 0.15 g for platinum. Concerning the deflector surface, this concentration corresponded to 140, 280, and 420 $g \cdot m^{-2}$ for $CuO$, $TiO_2$, and $MnO_2$, and 0.4, 0.8, and 1.2 $g \cdot m^{-2}$ for platinum catalysts. All the catalysts used contributed to an increase in the combustion temperature and a reduction in pollutant emissions. The results presented in the paper will allow the implementation of the developed solutions in the industry producing low-power boilers and in already-existing heating installations. The factor that motivates the introduction of changes may be continuously tightening European emission regulations.

**Keywords:** combustion; solid catalysts; exhaust gas purifying; catalyst; biomass; low-power boilers

## 1. Introduction

The energy policy of the European Union (EU) is aimed at achieving three primary goals, i.e., minimizing the harmful consequences of the functioning of energy technologies, keeping energy prices at the lowest possible level, and ensuring the security of energy supply in the short and long term [1]. An essential element of the EU energy policy is striving to increase the efficiency of energy production processes. Actions to increase competition in energy markets and develop international energy trade are also promoted [2]. According to the Green Book "Framework for climate and energy policy until 2030" published in 2013 by the European Commission, the promoted solution to the problem of increasing environmental degradation is decarbonizing the energy production sector and increasing the share of renewable energy sources [3]. The development directions of the energy sector set by the EU raise some concerns. Specialists emphasize the unstable nature of technologies for obtaining energy from renewable sources, which is the main problem of basing energy security on them. Energy security enforces the readiness of conventional

units, which leads to a reduction in the ecological aspects of the use of renewable energy [4]. As a result of changes in the energy market and the displacement of stable conventional sources, an increase in energy prices is recorded. There are also cases of the negative impact of renewable energy on the functioning of the power system. A good solution seems to be the increase in the share of biomass use in energy conversion processes. This will allow the stability of a conventional system to be maintained while achieving some of the assumptions of the EU energy policy relating to environmental protection [5].

As an EU member state, Poland is also starting to change its model of the energy generation system, as evidenced by the so-called state strategic documents. The most important of them is the National Energy and Climate Plan for the years 2021–2030 and Poland's Energy Policy until 2040. Both plans describe the directions of the renewable energy market development [6]. The Polish Energy Policy until 2040, adopted on 2 February 2021, is mainly based on the following assumptions: improving energy efficiency, increasing the security of fuel and energy supplies, developing the use of renewable energy sources, including biofuels, and reducing the impact of energy on the environment [7]. Biomass is recognized as a significant renewable energy source in each legal act. The regulations consider this source's great potential, mainly agricultural and forestry biomass [8].

It is estimated that $1.4 \cdot 10^{11}$ MJ of energy from biomass is currently used to heat single-family houses in Poland. This value is growing every year. The current trend of abandoning the most popular energy raw material, hard coal, favors the development of techniques for generating energy from biomass [9]. The market of producers of biomass boilers in Poland is constantly growing, resulting from the doubling of the number of biomass boilers sold each year. The development prospects of this sector are supported by the amendment to the technical conditions (WT. 2021) for newly constructed buildings, which entered into force in 2021 [10]. Generating heat from biomass will be a cheaper solution than its equivalents based on a different RES source. Such a situation will lead to the further dynamic development of the heat generation technology from biomass due to its positive economic and environmental aspects [11].

Air pollution is defined as any substance that may be harmful to humans, animals, and plants. The scale of the process, type, and properties of the fuel used significantly impact the emission of pollutants from biomass combustion [12]. During the combustion process, incomplete combustion products and products associated more closely with the fuel's physicochemical properties than with the combustion process conditions appear [13]. The products of incomplete combustion include carbon monoxide, hydrocarbons, tar, aromatic hydrocarbons, and unburned fuel particles. These substances are usually formed due to too low combustion temperature, insufficient mixing of fuel particles with air, or too-short residence time of the reactants in the high-temperature zone [14]. Usually, lower emissions of harmful substances contained in exhaust gases are achieved during combustion with higher efficiency [13]. Another group of combustion products consists of compounds whose formation is unrelated to the combustion process's incomplete course. Their emission is mainly related to the fuel's chemical composition and physical properties. In this case, stoichiometry and the process conditions have a more negligible influence on the content of harmful substances in the exhaust gas. As a result of the combustion process, solid particles, nitrogen and sulfur oxides, hydrogen chloride, and heavy metals are emitted [13].

The concept of catalysis was introduced by Berzelius in 1836 to explain the course of the decomposition and transformation reactions [15,16]. The main task of the catalyst is to reduce the activation energy, shortening the time it takes for the system to reach equilibrium. The change in energy barriers for individual reactions may vary. One can change the selectivity by changing the speed of individual reactions [17].

Currently, various technologies are available on the market to reduce the emission of harmful substances. Part of the gas-cleaning installation is dedicated to a selected group of emitters, incl: power plants, internal combustion engines, and the chemical industry.

The remaining methods are more universal and can be applied to other emitters struggling with exceeded emission limits, mainly for nitrogen oxides [18]. Modification aimed at reducing the number of pollutants, including NOx, can be carried out when mixing the fuel with the oxidant, fuel ignition, during the combustion process, or in exhaust gas treatment installations [19]. There are three types of methods for reducing nitrogen oxide emissions, including modification at the stage preceding the combustion process, changes at the stage of the combustion process, and exhaust gas treatment [20]. Effective technology development is complicated due to the complexity of the optimized parameters. In addition to reducing NOx emissions, one should take into account, among other things, the total efficiency of the installation, variability of the properties of the fuel burned (especially for biomass), variability in the load on the installation, and limitation of the negative impact on the emission of other pollutants, e.g., CO [19]. Despite the availability of many NOx reduction technologies, there has been continuous development towards improving their efficiency and developing new methods. This is due to the inability to meet the increasingly stringent legal emission limitations [21,22].

The sources of pollutant emissions in Poland are varied, but the most harmful pollutants come from the municipal and housing sector. These emitters are called the low-emission sector [23]. Low emission is defined as the emission of harmful dust and gases from emitters not exceeding 40 m in height [24]. Such a situation leads to the accumulation of pollutants in the zone of permanent human residence [25]. Estimating emissions and reducing emissions is complex. It is related to the seasonality in the use of low-power boilers. In large cities, transport also contributes to low emissions. Currently, the industry has little influence on the phenomenon of low emission. The reason is the stringent regulations and standards that entrepreneurs must meet in order not to incur severe penalties for excessive emissions [26]. In order to minimize the impact of the main component of low emissions (the municipal and household sector), the state introduces legal regulations concerning the quality of fuel and the efficiency of heating devices. Following the requirements of the classification of boilers (fifth class) introduced in 2017, the emission of harmful substances, except for NOx, has been tightened and regulated. Until 2020 (the period of entry into force of the Ecodesign Directive), emissions of nitrogen oxides from low-power boilers were not included in the regulations. The Ecodesign Directive introduced a NOx emission limit of 200 mg·m-3 for biomass boilers. These limits forced boiler manufacturers to interfere with the combustion process or to use an appropriate flue gas cleaning system [27–29].

The structure of this paper is as follows: Section 1 includes information on the strategies to reduce dependence on fossil fuels, the global problem of air pollution, its causes, and how to counter it through the use of catalysts. Section 2 presents the methods and materials used to test the effectiveness of the application of permanent catalysts in a low-power biomass boiler. Section 3 contains the test results of the catalyst's influence on the emissions of individual air pollutants and the combustion temperature. Section 4 is a discussion summarizing the research outcomes. Section 5 contains conclusions and information on the further work of the research team.

## 2. Materials and Methods

### 2.1. Biomass Fuels

Three biomass fuels were used to power the boiler: softwood pellets, wheat straw pellets, and hemp pellets. Softwood pellets are a commercial product (Olimp, the producer of Barlinek, Kielce, Poland) and are the most popular biomass energy carrier. The remaining prepared pellets were made using an 11 kW pellet production line (Figure 1).

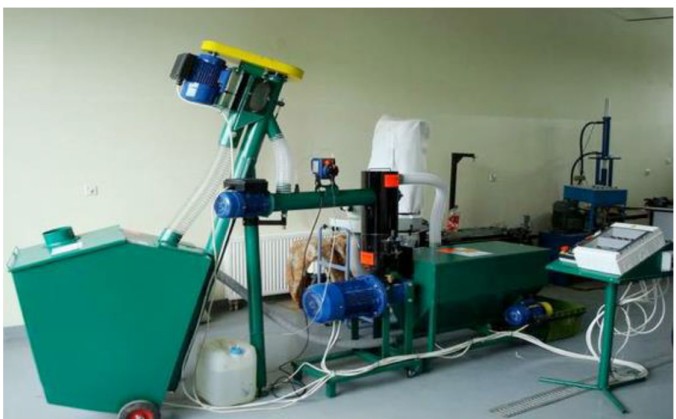

**Figure 1.** Kovo Novak pellet production line consisting of a hopper, granulation unit, and a cooler for finished pellets.

In the process of catalyst preparation, $TiO_2$ (War-Chem), $MnO_2$ (WarChem, Warsaw, Poland) oxides, Cu nitrate $(NO_3)_2 \times 3H_2O$ (WarChem, Warsaw, Poland), and $H_2PtC_{16}$ solution (Avantor Performance Materials Poland S.A., Gliwice, Poland ) were used. The oxides were applied to the surface of the carrier in the form of a previously prepared solution. All catalysts were prepared on a commercial aluminosilicate material used to fill the deflector.

### 2.2. Preparation of Catalysts

Four catalysts (copper, manganese, titanium, and platinum) were prepared. Each catalyst had three variants of the active substance concentration on the support surface: 17.5 g, 35 g, and 52.5 g for the prepared catalysts with CuO, $TiO_2$, $MnO_2$ and, respectively, 0.05 g, 0.1 g, and 0.15 g for platinum catalysts. Concerning the deflector surface, this concentration corresponded to 140, 280, and 420 g·m$^{-2}$ for CuO, $TiO_2$, and $MnO_2$ and 0.4, 0.8, and 1.2 g·m$^{-2}$ for platinum catalysts.

The copper catalyst (Figure 2) was prepared by spraying the copper (II) nitrate solution onto the surface of the boiler deflector. Spraying was carried out through a MAROLEX model RX spraying device with a full cone nozzle. A diaphragm pump with a pressure and flow regulator was installed in the device. After applying a predetermined amount of the solution to the deflector surface, the device turned off. In order to evaporate the moisture from the carrier, a laboratory dryer Wamed KBC-65W was used. Drying was carried out at a temperature of 110 °C for 24 h. The whole process was repeated until the appropriate concentration of copper particles on the support surface was obtained. Each subsequent catalyst was made according to the same procedure.

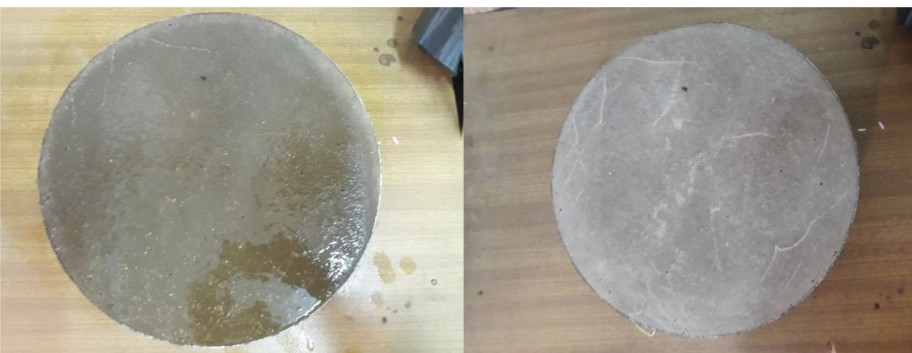

**Figure 2.** Catalyst containing copper particles shortly after spraying (**left**) and after the drying process (**right**).

*2.3. Biofuel Physicochemical Analysis*

The technical analysis of the biofuels used to power the boiler was carried out following the standards applicable to solid biofuels. The technical analysis included determining such fuel properties as moisture, ash, volatile matter content, and the gross and net calorific value. All tests in the field of analyzing the physicochemical properties of materials were carried out in triplicate.

The moisture content in the fuel was determined using the weighing–dryer method using a laboratory dryer (WAMED KBC-65W), following the PN-EN standard [30]. The technical specification of the used laboratory dryer is presented in Table 1.

**Table 1.** Technical specification of WAMED KBC-65W laboratory dryer.

| Parameter | Unit | Value |
|---|---|---|
| Capacity | dm³ | 75 |
| Rated power | W | 1500 |
| Continuous operating temperature | °C | +10–300 |
| Accuracy of temperature stabilization at a point along the center axis of the chamber | °C | ±0.2 |

The value of the combustion heat was determined using the IKA C 200 calorimeter. The calorific value was determined following the PN-EN standard [31]. The technical data of the used calorimeter are presented in Table 2.

**Table 2.** Technical specification of IKA C 200 calorimeter.

| Parameter | Unit | Value |
|---|---|---|
| Maximum output energy | J | 40,000 |
| Temperature sensor resolution | °C | 0.0001 |
| Working pressure of oxygen | bar | 40 |
| Initial temperature settings | °C | 18–25 |

The content of mineral parts in the fuel (content after ashing) was determined by soaking crucibles containing a certain amount of fuels in a muffle furnace (SNOL 8.2/1100) at a temperature of 550°C, following the requirements of the PN-EN standard [32]. According to the PN-EN standard, the fuel's volatile parts were determined by heating in an inert atmosphere in a muffle furnace [33]. The inert atmosphere in the muffle furnace chamber was obtained by introducing carbon dioxide during the analysis. The technical specification of the used muffle furnace is presented in Table 3.

**Table 3.** Technical specification of SNOL 8.2/1100 muffle furnace.

| Parameter | Unit | Value |
|---|---|---|
| Useful volume | dm³ | 8.2 |
| Rated power | W | 1800 |
| Continuous operating temperature | °C | +10–1100 |
| Temperature stability in working chamber at rated temperature in thermal steady state without charge not more than | °C | ±1 |

The elemental composition of biomass fuels, i.e., the determination of carbon, hydrogen, nitrogen, and sulfur, was performed using the PerkinElmer CHNS/O 2400 apparatus. The tests were carried out on samples in the dry state and crushed to a grain size smaller than 0.2 mm, following the requirements of the PN-EN standard [34]. The technical specification of the used apparatus is presented in Table 4.

**Table 4.** Technical specification of PerkinElmer CHNS/0 2400 apparatus.

| Parameter | Unit | Value |
|---|---|---|
| Temperature range | °C | 100–1100 |
| Sample size | mg | 0–500 |
| Accuracy | % | ≤0.3 |
| Carbon analytical range | mg | 0.001–3.6 |
| Hydrogen analytical range | mg | 0.001–1.0 |
| Nitrogen analytical range | mg | 0.001–6.0 |
| Sulphur analytical range | mg | 0.001–2.0 |
| Oxygen analytical range | mg | 0.001–2.0 |

*2.4. Research Stand*

The research was conducted in a fully automatic retort boiler with nominal thermal power of 15 kW (EG-PELLET 15, produced by EKOGREŃ, Pszczyna, Poland). A controller regulated its operating parameters. The temperature sensor readings and the lambda probe supplied the appropriate dose of fuel and air to the combustion chamber. An exhaust fan and a screw feeder coupled with a container provided the air dose and the fuel. The generated heat energy was transferred to the environment through fan heaters with a heating capacity of 40 kW. The catalytic system was placed in the initially installed boiler deflector. The location of the catalytic system is shown in the diagram of the combustion chamber structure (Figure 3). The figure also shows the forced flow of exhaust gases in the chamber.

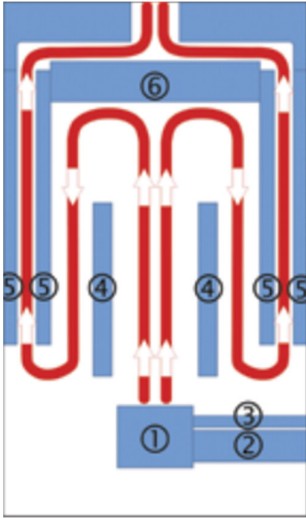

**Figure 3.** Scheme of the combustion chamber structure in cross section with flue gas flow: 1—burner, 2—pelet feeder, 3—aeration pipe, 4—partition, 5—water jacket, 6—exhaust deflector (location of the catalyst).

*2.5. Analysis of Exhaust Gas Composition*

The measurement was performed using the Testo 350 flue gas analyzer. The detection of individual compounds in the exhaust gas was performed based on the photochemical method. The exhaust gas composition recording was started after combustion process stabilization. The measurement lasted 5 h continuously, and the results were recorded every 1 s.

The used analyzer technical data are shown in Table 5.

**Table 5.** Technical specification of the Testo 350 flue gas analyzer.

| Component | Measurement Method | Range | Precision | Compl. with Standards |
|-----------|--------------------|-------|-----------|------------------------|
| $O_2$ | paramagnetic | 0–25% | ± 0.1% abs. or 3% rel. | EN 14789; OTM-13 |
| CO | chemiluminescence | 0–10,000 ppm | ± 3 ppm abs. or 3% rel. | EN 15058; METHOD 10 |
| $CO_2$ | chemiluminescence | 0–25% | ± 0.03% abs. or 3% rel. | ISO 12039; OTM-13 |
| $NO_x$ | chemiluminescence | 0–1000 ppm | ± 3 ppm abs. or 3% rel. | EN14792 |
| $SO_2$ | chemiluminescence | 0–800 ppm | ± 5 ppm abs. or 5% rel. | EN14793 |

The particulate matter content in the flue gas was measured using a Testo 380 particulate matter analyzer. It was determined as a suspended dust sum without division into individual fractions. The amount of PM in the exhaust gas was measured based on the infrared detection method. Prior to the measurement, the apparatus was conditioned.

The specification of used particulate matter analyzer is shown in Table 6.

**Table 6.** Technical data of the Testo 380 analyzer.

| Component | Measurement Method | Range | Precision | Compl. with Standards |
|-----------|--------------------|-------|-----------|------------------------|
| PM | NDIR | 0–300 mg·m$^{-3}$ | ± 1 ppm abs. or 1% rel. | EN14842 |
| $CO_2$ | chemiluminescence | 0–20% | ± 0.03% abs. or 3% rel. | ISO 12039 |
| $O_2$ | paramagnetic | 0–22% | ± 0.1% abs. or 3% rel. | EN 14789 |

The analyzer and dust meter probe were placed through a stub pipe in the chimney at a distance of 30 cm from the boiler flue gas outlet. Aromatic hydrocarbons (PAHs) and volatile organic compounds (VOC) samples were taken using an aspiration system with adsorption on activated carbon (SKC LAT 120) and a hydrophobic organic polymer (XAD-2). The system consisted of two measurement paths (independent for VOC and PAH). Each track included an aspiration probe tightly mounted in the stub pipe stub, filter, condenser, sorbent tubes (with active carbon for VOC and XAD-2 for PAH), and an aspirator (ASP-3 II). The aspiration probe was mounted on the axis of the conduit. For the first track (VOC), the gas consumption flow was 60 l·h$^{-1}$, and for the second track (PAH), 70 l·h$^{-1}$. The sampling time for the first track was 1 h and for the second track 5 h according to the standard [35]. After sampling, the sorbents were secured and transported to the laboratory. In laboratory conditions, desorption was carried out with carbon disulfide to perform a quantitative and qualitative analysis of PAHs and VOCs using a Varian 450GC gas chromatograph with a flame ionization detector.

*2.6. Measurement of the Temperature in the Combustion Chamber*

The temperature measurements in the combustion chamber were carried out using an APAR AR205 data recorder connected to four K-type thermocouples. They were placed in special connectors passing through the water jacket directly to the combustion chamber. The thermocouples were arranged parallel to the deflector and burner. The instantaneous temperature values were recorded with the frequency of 1 s for the entire measurement duration. Based on obtained results, the average values were determined. The outcomes included the mean measurement error, amounting to ± 1.5 °C.

## 3. Results

### 3.1. Biofuels Physicochemical Analysis

All biomass materials used as energy carriers were analyzed for their physicochemical properties. The basic technical properties of the analyzed biomass fuels are presented in Table 7.

**Table 7.** Results of technical analysis of biomass fuels.

| Parameter | Unit | Value for Wood Pellets | Value for Wheat Straw Pellets | Values for Hemp Pellets |
|---|---|---|---|---|
| Nitrogen content | % | 8.65 | 9.71 | 9.83 |
| Carbon content | % | 1.89 | 2.83 | 7.26 |
| Hydrogen content | % | 75.15 | 71.42 | 66.16 |
| Sulfur content | % | 18.51 | 15.96 | 17.01 |

The tested fuels are characterized by a relatively low moisture content, which directly affects the calorific value of the fuel. All pellets had few incineration residues, which proves the lack of significant mineral impurities in the fuel and forecasts lower operating loads. All fuels were characterized by a high content of volatile parts, which influenced their ignition temperature. The calorific value of the tested biomass was slightly differentiated. It was caused, among others, by different content of the carbon element in the elemental composition of fuels. This element is the main energy carrier in the mass of biomass, the content of which directly affects its calorific value. The ash content in the tested biomass is closely related to the mineral ballast content. Mineral particles and their concentration depend on the method of cultivating the plant and the agrotechnical procedures that they undergo. Therefore, hemp pellets have the highest concentration of mineral substances, which is also reflected in the value obtained in the incineration test. Therefore, the energy use of the tested fuels is justified, and the fuels made from biomass waste have parameters similar to the most popular energy carrier, which is wood pellet.

Biomass pellets used to power a low-power boiler during its operation were subjected to a basic analysis of elemental composition. The results obtained during the tests are presented in Table 8.

**Table 8.** Biomass fuels' elemental analysis results.

| Parameter | Unit | Value for Wood pellets | Value for Wheat Straw Pellets | Values for Hemp Pellets |
|---|---|---|---|---|
| Nitrogen content | % | 0.08 | 0.58 | 4.90 |
| Carbon content | % | 48.00 | 44.00 | 46.00 |
| Hydrogen content | % | 5.70 | 6.20 | 6.80 |
| Sulfur content | % | 0.02 | 0.07 | 0.27 |

The carbon content in the tested biomass fuels is similar and amounts to 48% for wood pellets, 46% for hemp pellets, and 44% for wheat straw pellets. The element whose content significantly affects the emission of its compounds is sulfur. The sulfur content in the fuels was small and amounted to 0.02% for wood pellets, 0.068% for straw pellets, and 0.27% for hemp pellets. The nitrogen content in the tested fuels is the most diverse and amounts to 0.08% for wood pellets, 0.58% for straw pellets, and 4.90% for hemp pellets. Considering the dominance of the fuel mechanism of nitrogen oxide formation during the combustion of biomass fuels in low-power boilers, individual contents will be reflected in the emission of toxins.

### 3.2. Measurement of the Temperature in the Combustion Chamber

During the analysis, the catalysts in the third (highest) active substance concentration turned out to be the most effective in reducing the number of pollutants. Therefore, in the further part of the article, only the results obtained for this most effective variant of catalysts are presented.

In the course of the conducted works, the temperature distribution in the combustion chamber of a low-power boiler was examined in all measurement cycles. Figure 4 shows a comparison of the average temperatures in the combustion chamber.

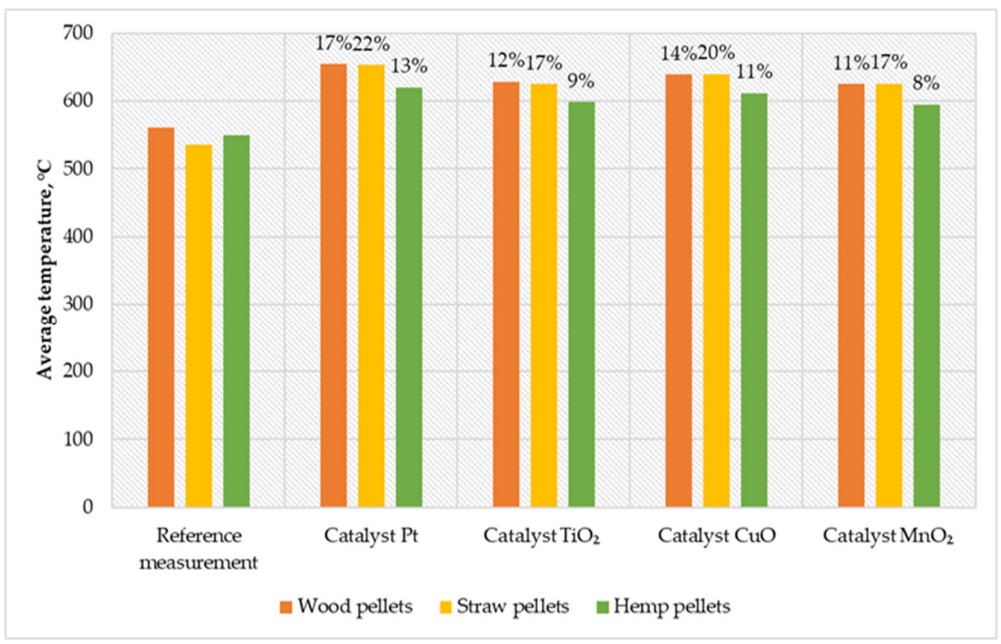

**Figure 4.** Comparison of average temperatures in the combustion chamber of a low-power biomass boiler.

During the measurement cycle, the highest average temperature in the chamber was recorded, equal to 653°C (for straw pellets with Pt catalyst). The catalyst variant based on platinum particles achieved the highest temperature increase in relation to the reference measurements (17% for wood pellets and 13% for hemp pellets). The catalyst with an active layer of copper oxide was characterized by an 11% effect on the temperature increase (for hemp pellets) and 20% (for straw pellets). The $TiO_2$ catalyst increased the temperature in relation to the reference measurements at 48°C (for hemp pellets) and 90°C (for straw pellets). Comparable results were obtained using the $MnO_2$ catalyst, where the growth rate for burning straw pellets was 17%, wood pellets 11%, and hemp pellets 8%.

### 3.3. Analysis of Exhaust Gas Composition

During all tests, the concentrations of individual components of exhaust gases were recorded. All concentrations were averaged and converted according to the adopted methodology to 10% oxygen content in the exhaust gas to standardize the emission levels. Figures 5–9 show the correlation between the concentration of combustion products in the exhaust gas and the type of the used catalytic system.

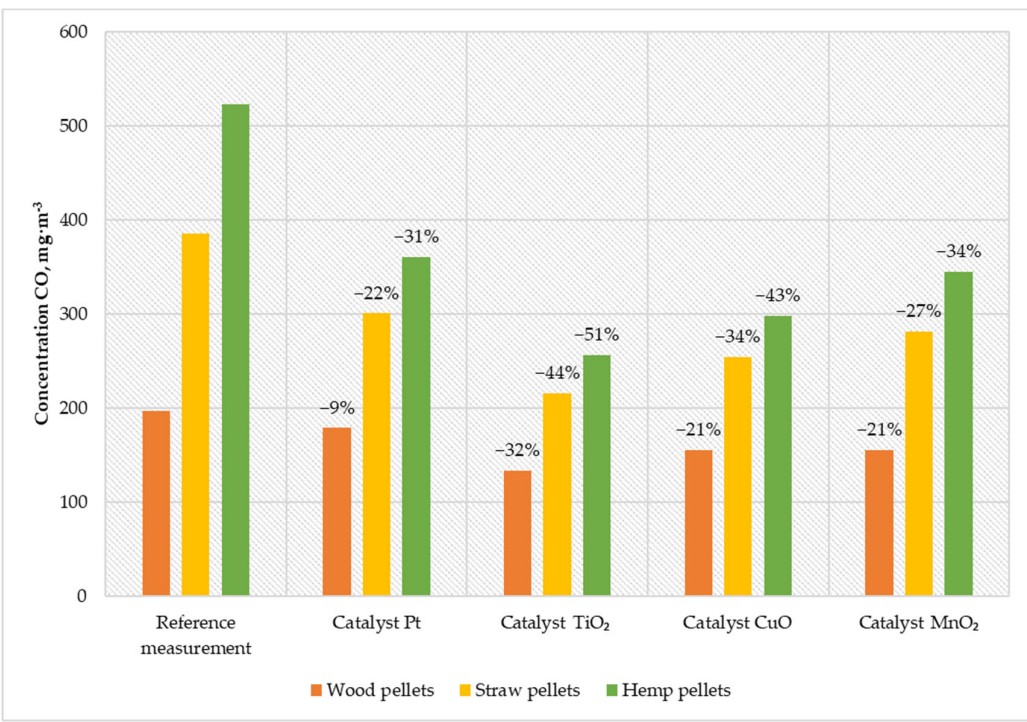

**Figure 5.** Average CO content in the flue gas from a low-power biomass boiler.

Thanks to installing a titanium-oxide-based catalyst, it was possible to reduce the CO content in the exhaust gases by more than half (in the case of burning hemp pellets). During the operation of the boiler with the installed CuO catalyst, a reduction in the carbon monoxide content of 21–43% was achieved (for wood pellets and hemp pellets, respectively). Using a catalyst based on manganese particles, the CO content in the exhaust gases was reduced by 42 $mg \cdot m^{-3}$ for wood pellets, 104 $mg \cdot m^{-3}$ for straw pellets, and 178 $mg \cdot m^{-3}$ for hemp pellets. The solution based on platinum nanoparticles was characterized by the lowest activity in relation to other catalysts. This catalyst allowed the reduction of carbon monoxide emissions from 197 $mg \cdot m^{-3}$ to 179 $mg \cdot m^{-3}$ for wood pellets, from 386 $mg \cdot m^{-3}$ to 301 $mg \cdot m^{-3}$ for straw pellets, and from 523 $mg \cdot m^{-3}$ to 361 $mg \cdot m^{-3}$ for hemp pellets.

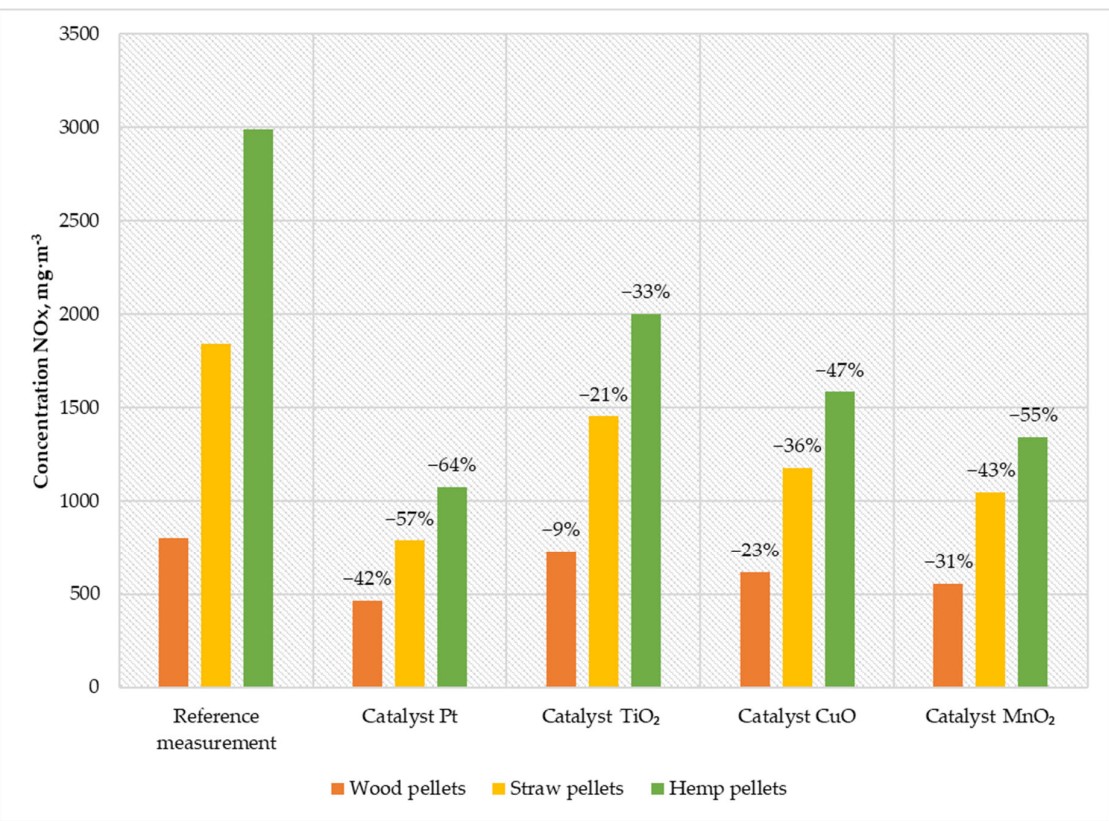

**Figure 6.** Average NO$_x$ content in the flue gas from a low-power biomass boiler.

The platinum catalyst was characterized by the highest reduction in nitrogen oxides, which was 42% for wood pellets, 57% for straw pellets, and 64% for hemp pellets. The second solution in terms of efficiency in reducing the amount of NO$_x$ in the exhaust gas was a catalyst based on manganese oxide. The reduction rate achieved was between 31% for wood pellets and 55% for hemp pellets. Using a copper catalyst reduced the emission of nitrogen oxides by an average of 185 mg·m$^{-3}$ for wood pellets, 664 mg·m$^{-3}$ for straw pellets, and 1406 mg·m$^{-3}$ for hemp pellets. The least favorable compared to other solutions was the TiO$_2$ catalyst, whose average reduction in nitrogen oxides in exhaust gases was between 9% for wood pellets and 33% for hemp pellets.

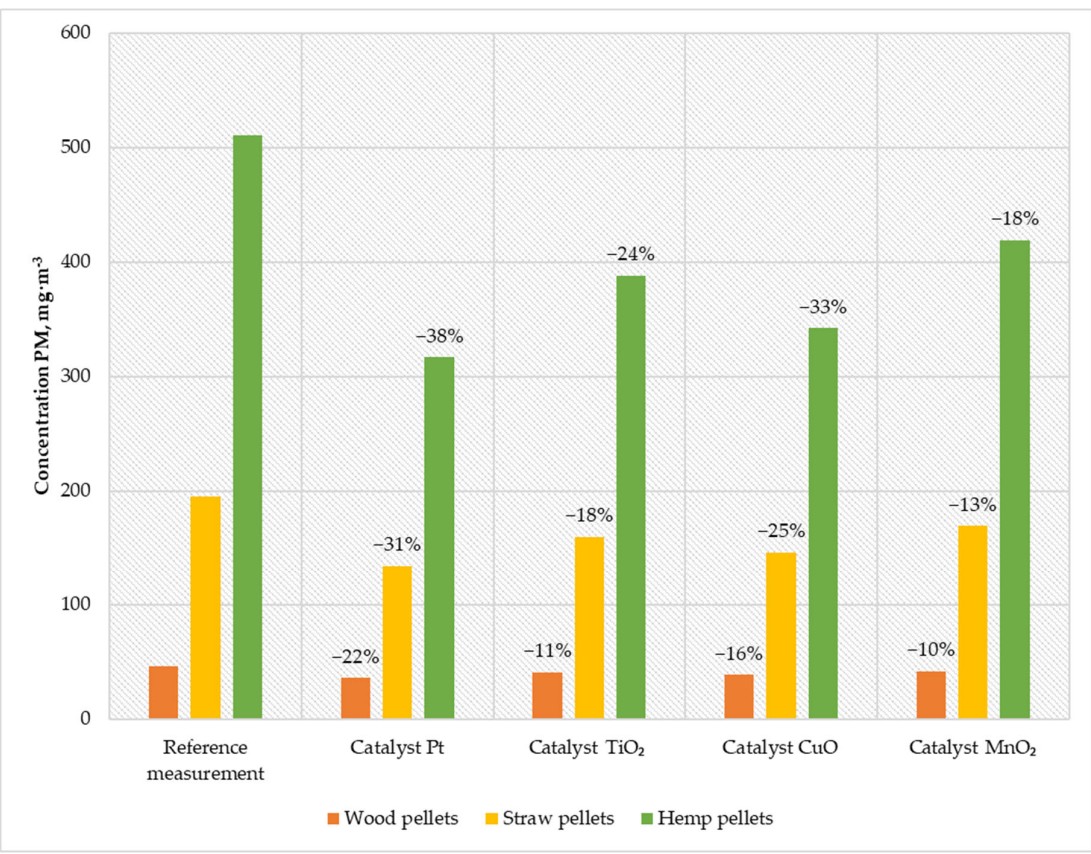

**Figure 7.** Average PM content in the flue gas from a low-power biomass boiler.

When feeding the boiler with wood pellets, the emission was 36 mg·m⁻³,;when burning straw pellets, the emission was 134 mg·m⁻³; and when using hemp pellets, the emission was 316 mg·m⁻³. Using the CuO catalyst, the emission was reduced to 38 mg·m⁻³ for wood pellets (meeting the requirements of the Ecodesign directive), 146 mg·m⁻³ for straw pellets, and 342 mg·m⁻³ for hemp pellets. Using a titanium catalyst caused PM emission at a level of approx. 40 mg·m⁻³ for wood pellets; while burning straw pellets, the emission was 159 mg·m⁻³; and when feeding the boiler with hemp pellets, the emission was 388 mg·m⁻³. The least effective solution was the MnO₂ catalyst, reducing suspended dust emissions by 10% for wood pellets, 13% for straw pellets, and 18% for hemp pellets.

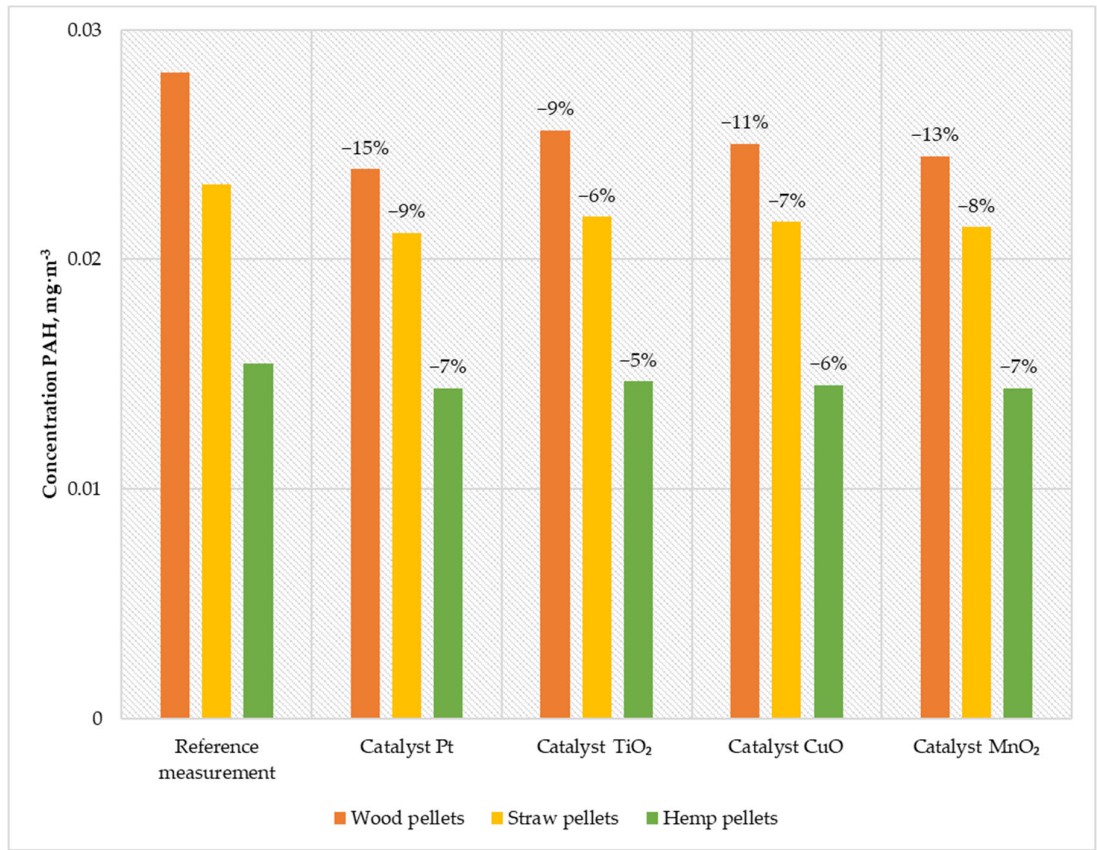

**Figure 8.** Average PAHs content in the flue gas from a low-power biomass boiler.

The results obtained during the boiler operation with the developed catalysts proved that the most effective solution for reducing PAHs is using a platinum catalyst. The emission of PAH during its use and feeding the boiler with wood pellets was 0.023 mg·m$^{-3}$. Feeding the boiler with straw pellets, the emission remained at the level of 0.021 mg·m-3, while during the combustion of hemp pellets, the emission was 0.014 mg·m$^{-3}$. The second most efficient solution is the manganese catalyst. When burning wood pellets, the emission was reduced to 0.024 mg·m$^{-3}$, burning straw pellets to 0.021 mg·m$^{-3}$, and burning hemp pellets to 0.014 mg·m$^{-3}$. The solution based on copper oxide was characterized by efficiency at the level of 11% when burning wood pellets, 7% when burning straw pellets, and 6% when burning hemp pellets. After installing the titanium catalyst, the PAH emissions generated during the combustion of wood pellets decreased by 9% while burning straw pellets by 6% and burning hemp pellets by 5%.

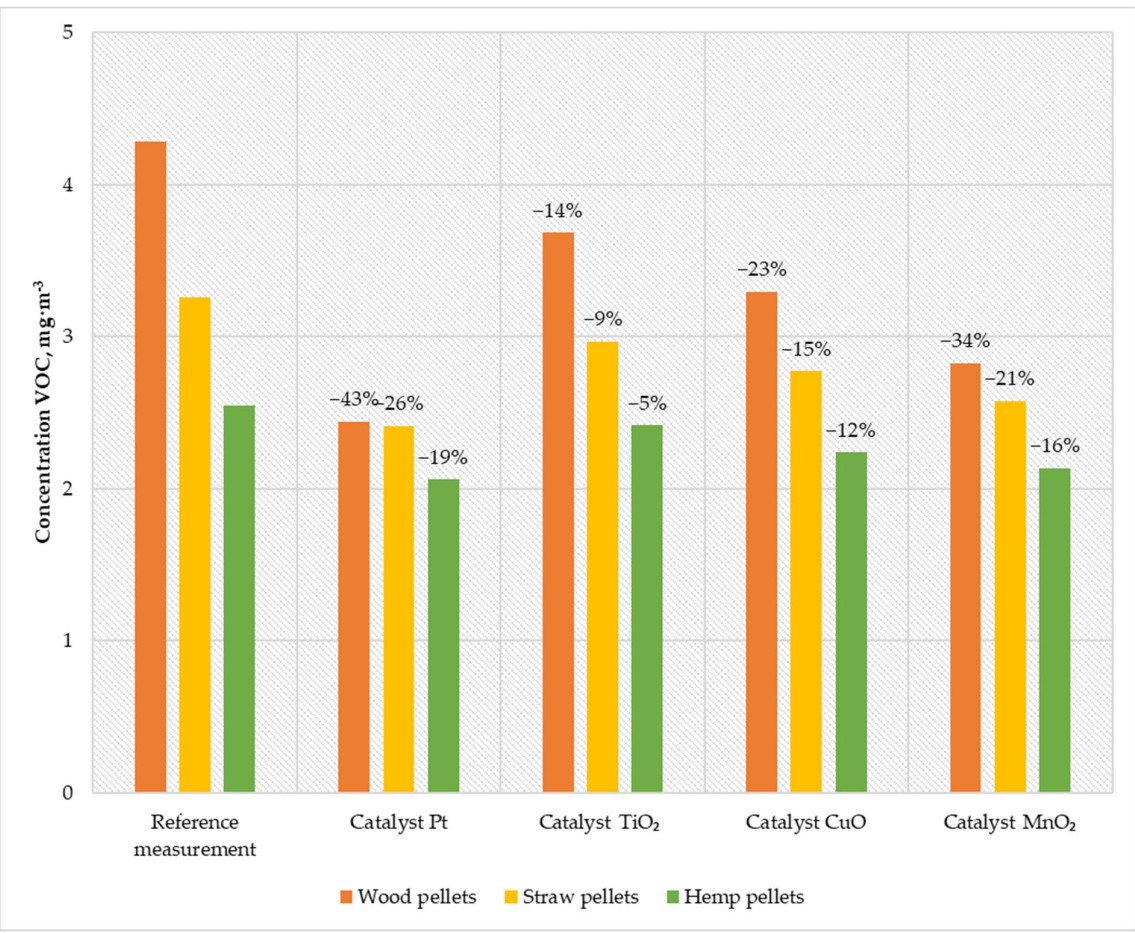

**Figure 9.** Average VOC content in the flue gas from a low-power biomass boiler.

The research carried out using prepared catalysts allowed obtaining high efficiency in reducing the amount of VOC in the exhaust gas. The use of the Pt catalyst reduced the emission generated during the combustion of wood pellets to 2.44 mg·m$^{-3}$; while burning straw pellets, the emission value decreased to 2.41 mg·m$^{-3}$; and when burning hemp pellets, the VOC concentration decreased to 2.06 mg·m$^{-3}$. Using the MnO$_2$ catalyst, the VOC emissions were 2.82 mg·m$^{-3}$ (for wood pellets), 2.57 mg·m$^{-3}$ (for straw pellets), and 2.13 mg·m$^{-3}$ (for hemp pellets). The variant of the catalyst based on CuO caused a decrease in the concentration of VOC in the flue gas to 3.29 mg·m$^{-3}$ when burning wood pellets, 2.77 mg·m$^{-3}$ by burning straw pellets, and 2.23 mg·m$^{-3}$ by feeding the boiler with hemp pellets. The TiO$_2$ catalyst was characterized by an efficiency of 5–14%, which allowed achieving emissions of 3.68 mg·m$^{-3}$ (wood pellets), 2.96 mg·m$^{-3}$ (straw pellets), and 2.41 mg·m$^{-3}$ (hemp pellets).

## 4. Discussion

During the research, it was observed that all the catalytic systems used increased the average temperature in the combustion chamber of a low-power boiler. The highest increment value was recorded for straw pellets with the variant of the Pt catalyst (118°C). The obtained values indicate that some changes occur from the thermodynamic point of view. The increase in temperature may be caused by the afterburning of flammable gases that are part of the exhaust gases, e.g., CO and soot, as well as the burning of fuel particles included in the drift, which previously went to the atmosphere with the exhaust gases. The catalysts used significantly increase the boiler's efficiency and reduce fuel consumption, resulting in an ecological and economic profit.

The presented test results confirm the ability of the tested catalysts to oxidize carbon monoxide. The catalyst $TiO_2$ and $CuO$ turned out to be the most effective solution. The oxidizing properties of these substances confirm this tendency. Nevertheless, any solution would meet the limit of the Ecodesign Directive (500 mg·m$^{-3}$). The use of catalysts improves the combustion process, increases the boiler efficiency, and reduces the losses resulting from the emission of the combustible gas, i.e., carbon monoxide. Reducing the CO content in the exhaust gas is economic and ecological because the main product resulting from the oxidation of carbon monoxide is $CO_2$, an inert gas, to the environment.

The analysis of the results obtained during the tests of various catalyst variants confirmed reducing the NOx content in the exhaust gas with their help. The results show that the catalyst based on platinum nanoparticles in the third concentration was the most effective solution. Thanks to this solution, it is possible to reduce the NOx content in the flue gas by 337 mg·m$^{-3}$ when burning wood pellets, by 1049 mg·m$^{-3}$ when burning straw pellets, and by 1915 mg·m$^{-3}$ when burning hemp pellets. The differences in the reduction rates for individual fuels resulted from the concentration of nitrogen oxides in the exhaust gases during combustion. Considering the reference emission level and the resultant one, approximately the overall reduction ratio remained very high (60–70%). Despite the obtained levels of reduction in NOx content in the exhaust gas, the threshold level included in the Ecodesign Directive (200 mg·m$^{-3}$) was not reached. This is because the exhaust gases that come into contact with the catalytic surface are too small. In order to bring the exhaust gas content to the normative level, catalytic surfaces should be applied proportionally to a more significant number of elements of the combustion chamber, or a combination of several types of catalysts should be used. Such solutions already require significant interference in the construction of the boiler.

The obtained results from all measurement cycles of the total concentration of suspended dust proved the positive effect of catalysts. The highest efficiency was achieved using a platinum catalyst (reduction at the level of 22–38% depending on the fuel used). The higher efficiency of the catalyst's operation was related to its higher operating temperature. Thanks to the catalytic systems, fuel particles in the lift, which previously ended up in the chimney, were burned off. Using a catalyst also reduces the problem of soot formation, which settles on the boiler's heating surfaces. Reducing the amount of dust in the flue gas is part of the desired ecological and economic effect, and, in the case of burning wood pellets, it allowed meeting the requirements of the Ecodesign standard (40 mg·m$^{-3}$).

The research proved that using catalysts positively affects the quality of exhaust gases. The maximum efficiency in reducing the amount of PAH achieved was 15% (for the Pt catalyst). Even a small degree of efficiency with such dangerous pollution significantly improves environmental profit.

The presented results confirmed the positive impact of catalysts on reducing VOC emissions in the flue gases from the low-power biomass boiler. The highest efficiency was achieved by using a catalyst based on platinum nanoparticles (19–43%). The VOC values in the flue gas in the boiler system without the catalyst oscillated below the applicable standard (20 mg·m$^{-3}$). Using all prepared catalysts for the combustion of wood pellets significantly reduced the amount of volatile organic compounds. The achieved catalyst temperature mainly influences this process. Solutions with a higher operating temperature have a more significant efficiency factor in reducing VOC.

## 5. Conclusions

By analyzing all the obtained results, it can be concluded that the metal catalysts developed by the research team installed in the original boiler deflector have a positive effect and improve the quality of exhaust gases. The applied catalytic systems caused a decrease in the concentrations of all measured pollutants, which gives a positive environmental effect. Solid biomass fuels are characterized by a high content of the nitrogen element.

According to the fuel cycle of nitrogen oxide formation in low-power boilers [36], the increased concentration of this element causes increased subsequent $NO_x$ emission in the flue gas. The best solution for reducing the amount of $NO_x$ in the exhaust gas turned out to be a catalyst based on platinum particles, which reached 64%. In order to reduce the emissions to the selected level (200 mg·m$^{-3}$), the active surface of the catalyst, which is in direct contact with the exhaust gas, should be increased, or the concentration of the active substance on its surface should be increased.

Considering the current market of energy resources and the pursuit of all economic sectors to trade in waste in a closed cycle, the energy use of biomass waste will become more and more widespread [37]. The only so-far-unsolved problem was the heterogeneity of the elemental composition and the increased emission of NOx, PM, PAH, and VOC generated during biomass combustion. The paper presents a way to solve this problem. The obtained results confirmed the reduction in harmful emissions in every examined aspect. Therefore, equipping boilers with the proposed catalytic systems will allow, to a greater extent, the use of biomass fuels produced from the hitherto undeveloped waste of the agro-food industry. Previously, researchers saw solutions to the problematic properties of biomass only in the process of pyrolysis or gasification [38]. However, these are energy-consuming and quite complicated processes, which bring tangible benefits only in the case of other materials, the properties of which do not allow for direct combustion. Therefore, biomass pyrolysis is currently studied mainly for the production of fertilizing materials [39].

The study proved many positive aspects resulting from the use of catalysts developed by the research team. However, in terms of nitrogen oxide emissions, the obtained results are unsatisfactory, as there was no decrease in emissions below the emissions allowed by the Ecodesign Directive. Therefore, the research team plans to conduct a follow-up study to refine the prepared catalytic systems. Further work will include the development of new, more effective active substances applied to catalytic supports and using inhibitors introduced into the combustion chamber together with the fuel or separately. Finally, a solution that can be implemented directly in producing low-power boilers will be created. This will allow the inclusion of a more significant amount of biomass waste in energy use and reduce emissions to the minimum safe level.

**Author Contributions:** Conceptualization, B.G. and B.K.; methodology, B.G. and B.K.; software, B.G. and B.K.; validation, B.G.; formal analysis, O.N.; investigation, B.G.; resources, B.G.; data curation, B.G.; writing—original draft preparation, B.G., B.K., M.J., O.N., T.N., and J.K.; writing—review and editing, B.G., B.K., and T.N.; visualization, B.G. and B.K.; supervision, B.G., B.K., O.N., and J.K. All authors have read and agreed to the published version of the manuscript.

**Funding:** The APC is financed by Wroclaw University of Environmental and Life Sciences.

**Institutional Review Board Statement:** Not applicable.

**Informed Consent Statement:** Not applicable.

**Data Availability Statement:** The data presented in this study are available on request from the corresponding author.

**Conflicts of Interest:** The authors declare no conflict of interest.

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
