# Peer review of "Influence of the Use of Permanent Catalytic Systems on the Flue Gases Emission from Biomass Low-Power Boilers"

_catalysts, doi:10.3390/catal12070710_

Round 1

Author Response

  1. The abstract was difficult to be understood due to the use of abbreviation, e.g. PM, PAH, and VOC, without any explanation.

As recommended by the reviewer, the abbreviations PAH, PM, and VOC were added to the abstract.

  1. The introduction is too long to find the main information that is directly relative to the content in present work.

As recommended by the reviewer, the introduction has been shortened and redrafted.

  1. The performance of four catalysts were investigated comparably, and pointed out that TiO2, CuO were distinct from MnO2 and Pt catalysts. But unfortunately, little explanation for their different performances could be found in the text.

In line with the reviewer's recommendation, information on the preparation of the remaining catalysts, the properties of individual active substances, and their impact on emissions was clarified and added.

  1. Similarly, three types of biomass pellets (wood pellets, wheat straw pellets, and hemp expeller) were also comparably studied. How to explain their different performances?

In line with the reviewer's recommendation, the differences in the physicochemical properties of the combusted biomass were explained.

Reviewer 2 Report

The authors are needed to improve the manuscript by following comments - 

1. The introduction is too long. The authors are needed to revise it.

2. Figure 1 should indicate different parts in the figure or in the caption.

3. For catalytical characterization there are no XRD or morphological (SEM or FESEM or TEM) results. The authors should be provided with these analyses.

4. The result and discussion parts should revise.

5. The conclusion is also too long. It should be more specific.

6. Overall, the whole manuscript should check the language through proofreading.

Author Response

1. The introduction is too long. The authors are needed to revise it.

As recommended by the reviewer, the introduction has been shortened and redrafted.

2. Figure 1 should indicate different parts in the figure or in the caption.

As recommended by the reviewer, the description of Figure 1 has been changed.

3. For catalytical characterization there are no XRD or morphological (SEM or FESEM or TEM) results. The authors should be provided with these analyses.

During extensive research, X-ray diffraction (XRD) analysis was performed using a Siemens D5000 diffractometer. The measurements were made for an angle of 2θ in the range of 5 ° -50 ° in 0.04 ° steps and 3 s per step. However, this work focuses on the effect of using catalysts on a technical scale. The results of the XRD analyses will be published in another manuscript focusing on the description of preparation methods and the study of reactivity in a specially designed reactor.

4. The result and discussion parts should revise.

In line with the reviewer's recommendation, the results and discussion have been redrafted and refined.

5. The conclusion is also too long. It should be more specific.

In line with the reviewer's recommendation, the conclusions were shortened and refined.

6. Overall, the whole manuscript should check the language through proofreading.

As recommended by the reviewer, the entire manuscript has been checked and revised in terms of linguistic correctness.

Round 2

Reviewer 1 Report

The manuscript has been well revised, so that it can be published as it present form.

Reviewer 2 Report

The authors revised the manuscript as per my previous comments. Therefore, I suggested accepting this paper.